# HAMILTONIAN GENERATIVE NETWORKS

**Peter Toth**[*]
DeepMind
petertoth@google.com

**Danilo J. Rezende**[*]
DeepMind
danilor@google.com

**Andrew Jaegle**
DeepMind
drewjaegle@google.com

**Sébastien Racanière**
DeepMind
sracaniere@google.com

**Aleksandar Botev**
DeepMind
botev@google.com

**Irina Higgins**
DeepMind
irinah@google.com

## ABSTRACT

The Hamiltonian formalism plays a central role in classical and quantum physics. Hamiltonians are the main tool for modelling the continuous time evolution of systems with conserved quantities, and they come equipped with many useful properties, like time reversibility and smooth interpolation in time. These properties are important for many machine learning problems —from sequence prediction to reinforcement learning and density modelling —but are not typically provided out of the box by standard tools such as recurrent neural networks. In this paper, we introduce the Hamiltonian Generative Network (HGN), the first approach capable of consistently learning Hamiltonian dynamics from high-dimensional observations (such as images) without restrictive domain assumptions. Once trained, we can use HGN to sample new trajectories, perform rollouts both forward and backward in time and even speed up or slow down the learned dynamics. [1] We demonstrate how a simple modification of the network architecture turns HGN into a powerful normalising flow model, called Neural Hamiltonian Flow (NHF), that uses Hamiltonian dynamics to model expressive densities. We hope that our work serves as a first practical demonstration of the value that the Hamiltonian formalism can bring to deep learning.

## 1 INTRODUCTION

Any system capable of a wide range of intelligent behaviours within a dynamic environment requires a good predictive model of the environment's dynamics. This is true for intelligence in both biological (Friston, 2009; 2010; Clark, 2013) and artificial (Hafner et al., 2019; Battaglia et al., 2013; Watter et al., 2015; Watters et al., 2019) systems. Predicting environmental dynamics is also of fundamental importance in physics, where Hamiltonian dynamics and the structure-preserving transformations it provides have been used to unify, categorise and discover new physical entities (Noether, 1915; Livio, 2012).

Hamilton's fundamental result was a system of two first-order differential equations that, in a stroke, unified the predictions made by prior Newtonian and Lagrangian mechanics (Hamilton, 1834). After well over a century of development, it has proven to be essential for parsimonious descriptions of nearly all of physics.

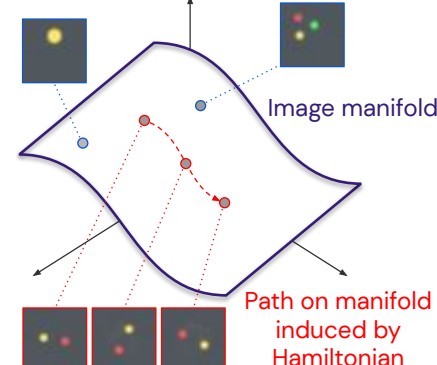

Figure 1: The Hamiltonian manifold hypothesis: natural images lie on a low-dimensional manifold in pixel space, and natural image sequences (such as one produced by watching a two-body system, as shown in red) correspond to movement on the manifold according to Hamiltonian dynamics.

---

[*]Equal contribution.

[1]More results and video evaluations are available at: http://tiny.cc/hgn

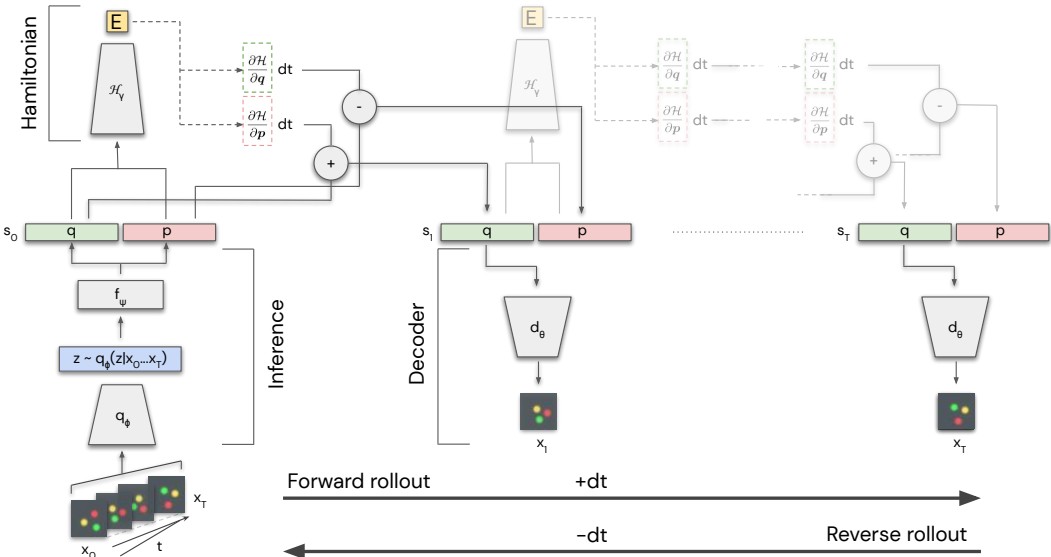

Figure 2: Hamiltonian Generative Network schematic. The encoder takes a stacked sequence of images and infers the posterior over the initial state. The state is rolled out using the learnt Hamiltonian. Note that we depict Euler updates of the state for schematic simplicity, while in practice this is done using a leapfrog integrator. For each unroll step we reconstruct the image from the position $q$ state variables only and calculate the reconstruction error.

Hamilton's equations provide a way to predict a system's future behavior from its current state in phase space (that is, its position and momentum for classical Newtonian systems, and its generalized position and momentum more broadly). Hamiltonian mechanics induce dynamics with several nice properties: they are smooth, they include paths along which certain physical quantities are conserved (symmetries) and their time evolution is fully reversible. These properties are also useful for machine learning systems. For example, capturing the time-reversible dynamics of the world state might be useful for agents attempting to account for how their actions led to effects in the world; recovering an abstract low-dimensional manifold with paths that conserve various properties is tightly connected to outstanding problems in representation learning (see e.g. Higgins et al. (2018) for more discussion); and the ability to conserve energy is related to expressive density modelling in generative approaches (Rezende & Mohamed, 2015). Hence, we propose a reformulation of the well-known image manifold hypothesis by extending it with a Hamiltonian assumption (illustrated in Fig. 1): natural images lie on a low-dimensional manifold embedded within a high-dimensional pixel space and *natural sequences of images trace out paths on this manifold that follow the equations of an abstract Hamiltonian.*

Given the rich set of established tools provided by Hamiltonian descriptions of system dynamics, can we adapt these to solve outstanding machine learning problems? When it comes to adapting the Hamiltonian formalism to contemporary machine learning, two questions need to be addressed: 1) how should a system's Hamiltonian be learned from data; and 2) how should a system's abstract phase space be inferred from the high-dimensional observations typically available to machine learning systems? Note that the inferred state may need to include information about properties that play no physical role in classical mechanics but which can still affect their behavior or function, like the colour or shape of an object. The first question was recently addressed by the Hamiltonian Neural Network (HNN) (Greydanus et al., 2019) approach, which was able to learn the Hamiltonian of three simple physical systems from noisy phase space observations. However, to address the second question, HNN makes assumptions that restrict it to Newtonian systems and appear to limit its ability to scale to more challenging video datasets.

In this paper we introduce the first model that answers both of these questions without relying on restrictive domain assumptions. Our model, the Hamiltonian Generative Network (HGN), is a generative model that infers the abstract state from pixels and then unrolls the learned Hamiltonian following the Hamiltonian equations (Goldstein, 1980). We demonstrate that HGN is able to reliably learn the Hamil-

tonian dynamics from noisy pixel observations on four simulated physical systems: a pendulum, a mass-spring and two- and three- body systems. Our approach outperforms HNN by a significant margin. After training, we demonstrate that HGN produces meaningful samples with reversible dynamics and that the speed of rollouts can be controlled by changing the time derivative of the integrator at test time. Finally, we show that a small modification of our architecture yields a flexible, normalising flow-based generative model that respects Hamiltonian dynamics. We show that this model, which we call Neural Hamiltonian Flow (NHF), inherits the beneficial properties of the Hamiltonian formalism (including volume preservation) and is capable of expressive density modelling, while offering computational benefits over standard flow-based models.

## 2 RELATED WORK

Most machine learning approaches to modeling dynamics use discrete time steps, which often results in an accumulation of the approximation errors when producing rollouts and, therefore, to a fast drop in accuracy. Our approach, on the other hand, does not discretise continuous dynamics and models them directly using the Hamiltonian differential equations, which leads to slower divergence for longer rollouts. The density model version of HGN (NHF) uses the Hamiltonian dynamics as normalising flows along with a numerical integrator, making our approach somewhat related to the recently published neural ODE work (Chen et al., 2018; Grathwohl et al., 2018). What makes our approach different is that Hamiltonian dynamics are both invertible and volume-preserving (as discussed in Sec. 3.3), which makes our approach computationally cheaper than the alternatives and more suitable as a model of physical systems and other processes that have these properties. Also related is recent work attempting to learn a model of physical system dynamics end-to-end from image sequences using an autoencoder (de Avila Belbute-Peres et al., 2018). Unlike our work, this model does not exploit Hamiltonian dynamics and is trained in a supervised or semi-supervised regime.

### 2.1 HAMILTONIAN NEURAL NETWORK

One of the most comparable approaches to ours is the Hamiltonian Neural Network (HNN) (Greydanus et al., 2019). This work, done concurrently to ours, proposes a way to learn Hamiltonian dynamics from data by training the gradients of a neural network (obtained by backpropagation) to match the time derivative of a target system in a supervised fashion. In particular, HNN learns a differentiable function $\mathcal{H}(q,p)$ that maps a system's state (its position, $q$, and momentum, $p$) to a scalar quantity interpreted as the system's Hamiltonian. This model is trained so that $H(p,q)$ satisfies the Hamiltonian equation by minimizing

$$\mathcal{L}_{\text{HNN}} = \frac{1}{2}[(\frac{\partial \mathcal{H}}{\partial p} - \frac{dq}{dt})^2 + (\frac{\partial \mathcal{H}}{\partial q} + \frac{dp}{dt})^2], \tag{1}$$

where the derivatives $\frac{\partial \mathcal{H}}{\partial q}$ and $\frac{\partial \mathcal{H}}{\partial p}$ are computed by backpropagation. Hence, this learning procedure is most directly applicable when the true state space (in canonical coordinates) and its time derivatives are known. Accordingly, in the majority of the experiments presented by the authors, the Hamiltonian was learned from the ground truth state space directly, rather than from pixel observations.

The single experiment with pixel observations required a modification of the model. First, the input to the model became a concatenated, flattened pair of images $\boldsymbol{o}_t = [\boldsymbol{x}_t, \boldsymbol{x}_{t+1}]$, which was then mapped to a low-dimensional embedding space $\boldsymbol{z}_t = [q_t, p_t]$ using an encoder neural network. Note that the dimensionality of this embedding ($z \in \mathbb{R}^2$ in the case of the pendulum system presented in the paper) was chosen to perfectly match the ground truth dimensionality of the phase space, which was assumed to be known a priori. This, however, is not always possible. The latent embedding was then treated as an estimate of the position and the momentum of the system depicted in the images, where the momentum was assumed to be equal to the velocity of the system – an assumption enforced by the additional constraint found necessary to encourage learning, which encouraged the time derivative of the position latent to equal the momentum latent using finite differences on the split latents:

$$\mathcal{L}_{\text{CC}} = (p_t - (q_{t+1} - q_t))^2. \tag{2}$$

This assumption is appropriate in the case of the simple pendulum system presented in the paper, however it does not hold more generally. Note that our approach does not make any assumptions on the dimensionality of the learned phase space, or the form of the momenta coordinates, which makes our approach more general and allows it to perform well on a wider range of image domains as presented in Sec. 4.

## 3 METHODS

### 3.1 THE HAMILTONIAN FORMALISM

The Hamiltonian formalism describes the continuous time evolution of a system in an abstract phase space $s = (q, p) \in \mathbb{R}^{2n}$, where $q \in \mathbb{R}^n$ is a vector of position coordinates, and $p \in \mathbb{R}^n$ is the corresponding vector of momenta. The time evolution of the system in phase space is given by the Hamiltonian equations:

$$\frac{\partial q}{\partial t} = \frac{\partial \mathcal{H}}{\partial p}, \quad \frac{\partial p}{\partial t} = -\frac{\partial \mathcal{H}}{\partial q} \tag{3}$$

where the Hamiltonian $\mathcal{H} : \mathbb{R}^{2n} \to \mathbb{R}$ maps the state $s = (q, p)$ to a scalar representing the energy of the system. The Hamiltonian specifies a vector field over the phase space that describes all possible dynamics of the system. For example, the Hamiltonian for an undamped mass-spring system is $\mathcal{H}(q, p) = \frac{1}{2} k q^2 + \frac{p^2}{2m}$, where $m$ is the mass, $q \in \mathbb{R}^1$ is its position, $p \in \mathbb{R}^1$ is its momentum and $k$ is the spring stiffness coefficient. The Hamiltonian can often be expressed as the sum of the kinetic $T$ and potential $V$ energies $\mathcal{H} = T(p) + V(q)$, as is the case for the mass-spring example. Identifying a system's Hamiltonian is in general a very difficult problem, requiring carefully instrumented experiments and researcher insight produced by years of training. In what follows, we describe a method for modeling a system's Hamiltonian from raw observations (such as pixels) by inferring a system's state with a generative model and rolling it out with the Hamiltonian equations.

### 3.2 LEARNING HAMILTONIANS WITH THE HAMILTONIAN GENERATIVE NETWORK

Our goal is to build a model that can learn a Hamiltonian from observations. We assume that the data $X = \{(x_0^1, ..., x_T^1), ..., (x_0^K, ..., x_T^K)\}$ comes in the form of high-dimensional noisy observations, where each $x_i = G(s_i) = G(q_i)$ is a non-deterministic function of the generalised position in the phase space, and the full state is a non-deterministic function of a sequence of images $s_i = (q_i, p_i) = D(x_0^i, ..., x_t^i)$, since the momentum (and hence the full state) cannot in general be recovered from a single observation. Our goal is to infer the abstract state and learn the Hamiltonian dynamics in phase space by observing $K$ motion sequences, discretised into $T + 1$ time steps each. In the process, we also want to learn an approximation to the generative process $G(s)$ in order to be able to move in both directions between the high dimensional observations and the low-dimensional abstract phase space.

Although the Hamiltonian formalism is general and does not depend on the form of the observations, we present our model in terms of visual observations, since many known physical Hamiltonian systems, like a mass-spring system, can be easily observed visually. In this section we introduce the Hamiltonian Generative Network (HGN), a generative model that is trained to behave according to the Hamiltonian dynamics in an abstract phase space learned from raw observations of image sequences. HGN consists of three parts (see Fig. 2): an inference network, a Hamiltonian network and a decoder network, which are discussed next.

**The inference network** takes in a sequence of images $(x_0^i, ...x_T^i)$, concatenated along the channel dimension, and outputs a posterior over the initial state $z \sim q_\phi(\cdot|x_0, ...x_T)$, corresponding to the system's coordinates in phase space at the first frame of the sequence. We parametrise $q_\phi(z)$ as a diagonal Gaussian with a unit Gaussian prior $p(z) = \mathcal{N}(0, \mathbb{I})$ and optimise it using the usual reparametrisation trick (Kingma & Welling, 2014). To increase the expressivity of the abstract phase space $s_0$, we map samples from the posterior with another function $s_0 = f_\psi(z)$ to obtain the system's initial state. As mentioned in Sec. 3.1, the Hamiltonian function expects the state to be of the form $s = (q, p)$, hence we initialise $s_0 \in \mathbb{R}^{2n}$ and arbitrarily assign the first half of the units to represent abstract position $q$ and the other half to represent abstract momentum $p$.

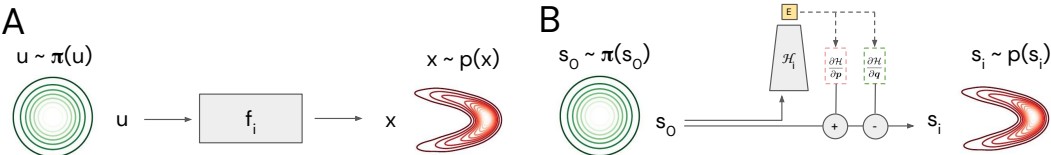

Figure 3: **A**: standard normalising flow, where the invertible function $f_i$ is implemented by a neural network. **B**: Hamiltonian flows, where the initial density is transformed using the learned Hamiltonian dynamics. Note that we depict Euler updates of the state for schematic simplicity, while in practice this is done using a leapfrog integrator.

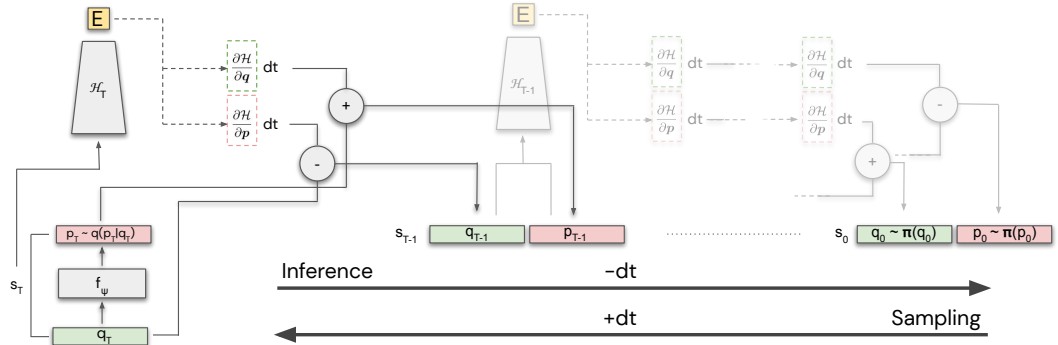

Figure 4: A schematic representation of NHF which can perform expressive density modelling by using the learned Hamiltonians as normalising flows. Note that we depict Euler updates of the state for schematic simplicity, while in practice this is done using a leapfrog integrator.

**The Hamiltonian network** is parametrised as a neural network with parameters $\gamma$ that takes in the inferred abstract state and maps it to a scalar $\mathcal{H}_\gamma(\boldsymbol{s}_t) \in \mathbb{R}$. We can use this function to do rollouts in the abstract state space using the Hamiltonian equations (Eq. 3), for example by Euler integration: $\boldsymbol{s}_{t+1} = (\boldsymbol{q}_{t+1}, \boldsymbol{p}_{t+1}) = (\boldsymbol{q}_t + \frac{\partial \mathcal{H}}{\partial \boldsymbol{p}_t} dt, \boldsymbol{p}_t - \frac{\partial \mathcal{H}}{\partial \boldsymbol{q}_t} dt)$. In this work we assume a separable Hamiltonian, so in practice we use a more sophisticated leapfrog integrator to roll out the system, since it has better theoretical properties and results in better performance in practice (see Sec. A.6 in Supplementary Materials for more details).

**The decoder network** is a standard deconvolutional network (we use the architecture from Karras et al. (2018)) that takes in a low-dimensional representation vector and produces a high-dimensional pixel reconstruction. Given that each instantaneous image does not depend on the momentum information, we restrict the decoder to take only the position coordinates of the abstract state as input: $p_\theta(\boldsymbol{x}_t) = d_\theta(\boldsymbol{q}_t)$.

**The objective function.** Given a sequence of $T+1$ images, HGN is trained to optimise the following objective:

$$\mathcal{L}(\phi, \psi, \gamma, \theta; \boldsymbol{x}_0, ... \boldsymbol{x}_T) = \frac{1}{T+1} \sum_{t=0}^{T} [\, \mathbb{E}_{q_\phi(\boldsymbol{z}|\boldsymbol{x}_1, ... \boldsymbol{x}_T)} [\, \log p_{\psi, \gamma, \theta}(\boldsymbol{x}_t \mid \boldsymbol{q}_t) \,] \,] - KL(\, q_\phi(\boldsymbol{z}) \,\|\, p(\boldsymbol{z}) \,),$$

(4)

which can be seen as a temporally extended variational autoencoder (VAE) (Kingma & Welling, 2014; Rezende et al., 2014) objective, consisting of a reconstruction term for each frame, and an additional term that encourages the inferred posterior to match a prior. The key difference with a standard VAE lies in how we generate rollouts – these are produced using the Hamiltonian equations of motion in learned Hamiltonian phase space.

## 3.3 LEARNING HAMILTONIAN FLOWS

In this section, we describe how the architecture described above can be modified to produce a model for flexible density estimation. Learning computationally feasible and accurate estimates of complex

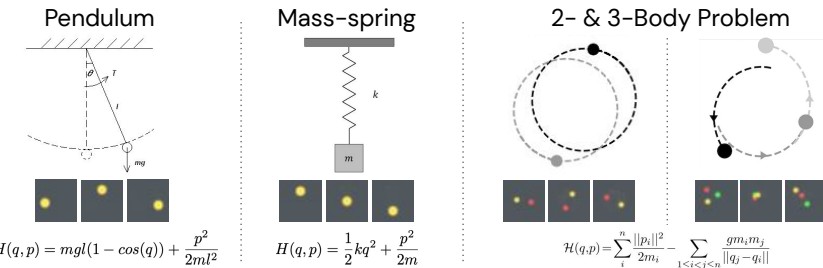

Figure 5: Ground truth Hamiltonians and samples from generated datasets for the ideal pendulum, mass-spring, and two- and three-body systems used to train HGN.

densities is an open problem in generative modelling. A common idea to address this problem is to start with a simple prior distribution $\pi(\boldsymbol{u})$ and then transform it into a more expressive form $p(\boldsymbol{x})$ through a series of composable invertible transformations $f_i(\boldsymbol{u})$ called normalising flows (Rezende & Mohamed, 2015) (see Fig. 3A). Sampling can then be done according to $\boldsymbol{x} = f_T \circ ... \circ f_1(\boldsymbol{u})$, where $\boldsymbol{u} \sim \pi(\cdot)$. Density evaluation, however, requires more expensive computations of both inverting the flows and calculating the determinants of their Jacobians. For a single flow step, this equates to the following: $p(\boldsymbol{x}) = \pi(f^{-1}(\boldsymbol{x}))\left|\det\left(\frac{\partial f^{-1}}{\partial \boldsymbol{x}}\right)\right|$. While a lot of work has been done recently into proposing better alternatives for flow-based generative models in machine learning (Rezende & Mohamed, 2015; Kingma et al., 2016; Papamakarios et al., 2017; Dinh et al., 2017; Huang et al., 2018; Kingma & Dhariwal, 2018; Hoogeboom et al., 2019; Chen et al., 2019; Grathwohl et al., 2018), none of the approaches manage to produce both sampling and density evaluation steps that are computationally scalable.

The two requirements for normalising flows are that they are invertible and volume preserving, which are exactly the two properties that Hamiltonian dynamics possess. This can be seen by computing the determinant of the Jacobian of the infinitesimal transformation induced by the Hamiltonian $\mathcal{H}$. By Jacobi's formula, the derivative of the determinant at the identity is the trace and:

$$\det\left[\mathbb{I} + dt\begin{pmatrix} \frac{\partial^2 \mathcal{H}}{\partial q_i \partial p_j} & -\frac{\partial^2 \mathcal{H}}{\partial q_i \partial q_j} \\ \frac{\partial^2 \mathcal{H}}{\partial p_i \partial p_j} & -\frac{\partial^2 \mathcal{H}}{\partial p_i \partial q_j} \end{pmatrix}\right] = 1 + dt\,\mathrm{Tr}\begin{pmatrix} \frac{\partial^2 \mathcal{H}}{\partial q_i \partial p_j} & -\frac{\partial^2 \mathcal{H}}{\partial q_i \partial q_j} \\ \frac{\partial^2 \mathcal{H}}{\partial p_i \partial p_j} & -\frac{\partial^2 \mathcal{H}}{\partial p_i \partial q_j} \end{pmatrix} + O(dt^2) = 1 + O(dt^2) \quad (5)$$

where $i \neq j$ are the off-diagonal entries of the determinant of the Jacobian. Hence, in this section we describe a simple modification of HGN that allows it to act as a normalising flow. We will refer to this modification as the Neural Hamiltonian Flow (NHF) model. First, we assume that the initial state $\boldsymbol{s}_0$ is a sample from a simple prior $\boldsymbol{s}_0 \sim \pi_0(\cdot)$. We then chain several Hamiltonians $\mathcal{H}_i$ to transform the sample to a new state $\boldsymbol{s}_T = \mathcal{H}_T \circ ... \circ \mathcal{H}_1(\boldsymbol{s}_0)$ which corresponds to a sample from the more expressive final density $\boldsymbol{s}_T \sim p(\boldsymbol{x})$ (see Fig. 3B for an illustration of a single Hamiltonian flow). Note that unlike HGN, where the Hamiltonian dynamics are shared across time steps (a single Hamiltonian is learned and its parameters are shared across time steps of a rollout), in NHF each step of the flow (corresponding to a single time step of a rollout) can be parametrised by a different Hamiltonian. The inverse of such a Hamiltonian flow can be easily obtained by replacing $dt$ by $-dt$ in the Hamiltonian equations and reversing the order of the transformations, $\boldsymbol{s}_0 = \mathcal{H}_1^{-dt} \circ ... \circ \mathcal{H}_T^{-dt}(\boldsymbol{s}_T)$ (we will use the appropriate $dt$ or $-dt$ superscript from now on to make the direction of integration of the Hamiltonian dynamics more explicit). The resulting density $p(\boldsymbol{s}_T)$ is given by the following equation:

$$\ln p(\boldsymbol{s}_T) = \ln \pi(\boldsymbol{s}_0) = \ln \pi(\mathcal{H}_1^{-dt} \circ ... \circ \mathcal{H}_T^{-dt}(\boldsymbol{s}_T)) + O(dt^2)$$

Our proposed NHF is more computationally efficient that many other flow-based approaches, because it does not require the expensive step of calculating the trace of the Jacobian. Hence, the NHF model constitutes a more structured form of a Neural ODE flow (Chen et al., 2018), but with a few notable differences: (i) The Hamiltonian ODE is volume-preserving, which makes the computation of log-likelihood cheaper than for a general ODE flow. (ii) General ODE flows are only invertible in the limit $dt \to 0$, whereas for some Hamiltonians we can use more complex integrators (like the symplectic leapfrog integrator described in Sec. A.6) that are both invertible and volume-preserving for any $dt > 0$. The structure $\boldsymbol{s} = (\boldsymbol{q}, \boldsymbol{p})$ on the state-space imposed by the Hamiltonian dynamics can be constraining from the point of view of density estimation. We choose to use the trick proposed in the Hamiltonian

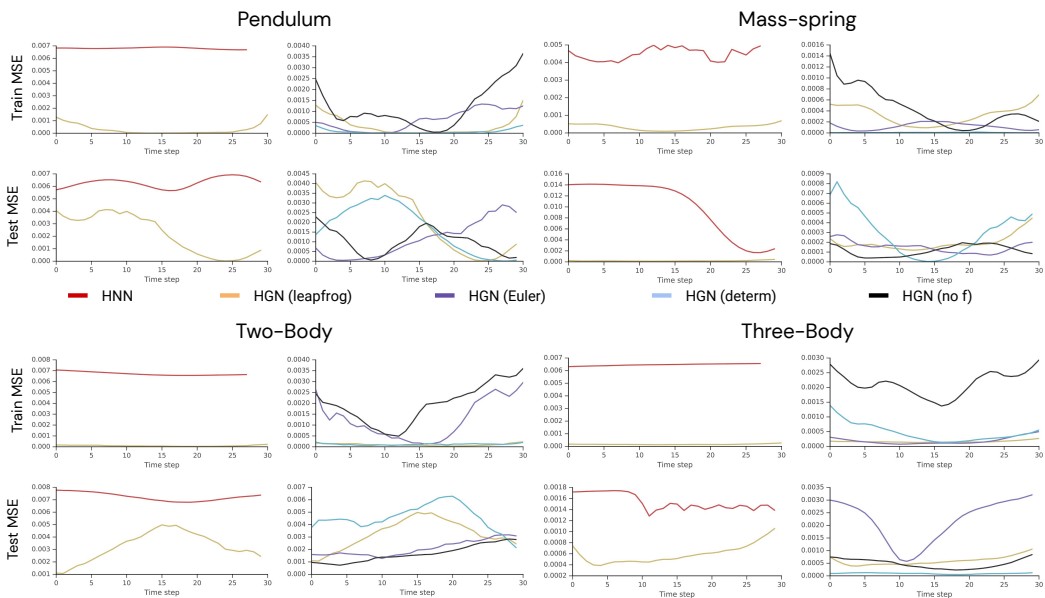

Figure 6: Average pixel MSE for each step of a single train and test unroll on four physical systems. All versions of HGN outperform HNN, which often learned to reconstruct an average image. Note the different scales of the plots comparing HGN to HNN and different versions of HGN.

Monte Carlo (HMC) literature (Neal et al., 2011; Salimans et al., 2015; Levy et al., 2017), which treats the momentum $\boldsymbol{p}$ as a latent variable (see Fig. 4). This is an elegant solution which avoids having to artificially split the density into two disjoint sets. As a result, the data density that our Hamiltonian flows are modelling becomes exclusively parametrised by $p(\boldsymbol{q}_T)$, which takes the following form: $p(\boldsymbol{q}_T) = \int p(\boldsymbol{q}_T, \boldsymbol{p}_T) d\boldsymbol{p}_T = \int \pi(\mathcal{H}_1^{-dt} \circ ... \circ \mathcal{H}_T^{-dt}(\boldsymbol{q}_T, \boldsymbol{p}_T)) d\boldsymbol{p}_T$. This integral is intractable, but the model can still be trained via variational methods where we introduce a variational density $f_\psi(\boldsymbol{p}_T|\boldsymbol{q}_T)$ with parameters $\psi$ and instead optimise the following ELBO:

$$\mathrm{ELBO}(\boldsymbol{q}_T) = \mathbb{E}_{f_\psi(\boldsymbol{p}_T|\boldsymbol{q}_T)}[\ln \pi(\mathcal{H}_1^{-dt} \circ ... \circ \mathcal{H}_T^{-dt}(\boldsymbol{q}_T, \boldsymbol{p}_T)) - \ln f_\psi(\boldsymbol{p}_T|\boldsymbol{q}_T)] \leq \ln p(\boldsymbol{q}_T), \quad (6)$$

Note that, in contrast to VAEs (Kingma & Welling, 2014; Rezende et al., 2014), the ELBO in Eq. 6 is not explicitly in the form of a reconstruction error term plus a KL term.

## 4 RESULTS

In order to directly compare the performance of HGN to that of its closest baseline, HNN, we generated four datasets analogous to the data used in Greydanus et al. (2019). The datasets contained observations of the time evolution of four physical systems: mass-spring, pendulum, two- and three-body (see Fig. 5). In order to generate each trajectory, we first randomly sampled an initial state, then produced a 30 step rollout following the ground truth Hamiltonian dynamics, before adding Gaussian noise with standard deviation $\sigma^2 = 0.1$ to each phase-space coordinate, and rendering a corresponding 64x64 pixel observation. We generated 50 000 train and 10 000 test trajectories for each dataset. When sampling initial states, we start by first sampling the total energy of the system denoted as a radius $r$ in the phase space, before sampling the initial state $(q,p)$ uniformly on the circle of radius $r$. Note that our pendulum dataset is more challenging than the one described in Greydanus et al. (2019), where the pendulum had a fixed radius and was initialized at a maximum angle of $30°$ from the central axis.

**Mass-spring.** The dynamics of a frictionless mass-spring system are modeled by the Hamiltonian $\mathcal{H} = \frac{1}{2}kq^2 + \frac{p^2}{2m}$, where $k$ is the spring constant and $m$ is the mass. We fix $k=2$ and $m=0.5$, then sample a radius from a uniform distribution $r \sim \mathbb{U}(0.1, 1.0)$.

**Pendulum.**    The dynamics of a frictionless pendulum are modeled by the Hamiltonian $\mathcal{H} = 2mgl(1 - \cos(q)) + \frac{p^2}{2ml^2}$, where $g$ is the gravitational constant and $l$ is the length of the pendulum. We fix $g = 3$, $m = 0.5$, $l = 1$, then sample a radius from a uniform distribution $r \sim \mathbb{U}(1.3, 2.3)$.

**Two- and three- body problems.**    In an n-body problem, particles interact with each other through an attractive force, like gravity. The dynamics are represented by the following Hamiltonian $\mathcal{H} = \sum_i^n \frac{||p_i||^2}{2m_i} - \sum_{1 \leq i < j \leq n} \frac{gm_i m_j}{||q_j - q_i||}$. We set $m = 1$ and $g = 1$ for both systems. For the two-body problem, we set $r \sim \mathbb{U}(0.5, 1.5)$, and we also change the observation noise to $\sigma^2 = 0.05$. For the three-body problem, we set $r \sim U(0.9, 1.2)$, and set the observation noise to $\sigma^2 = 0.2$.

**Learning the Hamiltonian**    We tested whether HGN and the HNN baseline could learn the dynamics of the four systems described above. To ensure that our re-implementation of HNN was correct, we replicated all the results presented in the original paper (Greydanus et al., 2019) by verifying that it could learn the dynamics of the mass-spring, pendulum and two-body systems well from the ground truth state, and the dynamics of a restricted pendulum from pixels. We also compared different modifications of HGN: a version trained and tested with an Euler rather than a leapfrog integrator (HGN Euler), a version trained with no additional function between the posterior and the prior (HGN no $f_\psi$) and a deterministic version (HGN determ), which did not include the sampling step from the posterior $q_\phi(\boldsymbol{z}|\boldsymbol{x}_0...\boldsymbol{x}_T)$.

| MODEL | MASS-SPRING | | PENDULUM | | TWO-BODY | | THREE-BODY | |
|---|---|---|---|---|---|---|---|---|
| | TRAIN | TEST | TRAIN | TEST | TRAIN | TEST | TRAIN | TEST |
| HNN | $50.38\pm1.75$ | $104.37\pm52.51$ | $69.39\pm0.89$ | $64.62\pm0.87$ | $80.52\pm1.79$ | $78.9\pm1.93$ | $61.63\pm3.31$ | $57.54\pm2.39$ |
| HNN (CONV) | $18.97\pm0.77$ | $119.09\pm69.17$ | $13.14\pm0.66$ | $106.07\pm41.94$ | $7.3\pm0.38$ | $120.71\pm44.48$ | $15.22\pm1.99$ | $62.77\pm37.57$ |
| HGN (EULER) | $3.67\pm1.09$ | $6.2\pm2.69$ | $5.43\pm2.53$ | $10.93\pm4.32$ | $6.62\pm3.93$ | $15.06\pm7.01$ | $7.51\pm3.49$ | $9.4\pm3.92$ |
| HGN (DETERM) | $0.23\pm0.23$ | $3.07\pm1.06$ | $0.79\pm1.24$ | $10.68\pm3.19$ | $2.34\pm2.3$ | $14.47\pm5.24$ | $4.1\pm2.05$ | $5.17\pm1.96$ |
| HGN (NO $f_\psi$) | $4.95\pm1.71$ | $7.04\pm2.55$ | $6.83\pm3.29$ | $13.98\pm4.94$ | $6.35\pm3.86$ | $16.49\pm6.6$ | $8.37\pm3.13$ | $10.41\pm3.72$ |
| HGN (LEAPFROG) | $3.84\pm1.07$ | $6.23\pm2.03$ | $4.9\pm1.86$ | $11.72\pm4.14$ | $6.36\pm3.29$ | $16.47\pm7.15$ | $7.88\pm3.55$ | $9.8\pm3.72$ |

Table 1: Average pixel MSE over a 30 step unroll on the train and test data on four physical systems. All values are multiplied by $1e+4$. We evaluate two versions of the Hamiltonian Neural Network (HNN) (Greydanus et al., 2019): the original architecture and a convolutional version closely matched to the architecture of HGN. We also compare four versions of our proposed Hamiltonian Generative Network (HGN): the full version, a version trained and tested with an Euler rather than a leapfrog integrator, a deterministic rather than a generative version, and a version of HGN with no extra network between the posterior and the initial state.

Tbl. 1 and Fig. 6 demonstrate that HGN and its modifications learned well on all four datasets. However, when we attempted to train HNN on the four datasets described above, its Hamiltonian often collapsed to 0 and the model failed to reproduce any dynamics, defaulting to a static single image. We were unable to improve on this performance despite our best efforts, including a modification of the architecture to closely match ours (referred to as HNN Conv) (see Sec. A.3 of the appendix for details). Tbl. 1 shows that the average mean squared error (MSE) of the pixel reconstructions on both the train and test data is an order of magnitude better for HGN compared to both versions of HNN. The same holds when visualising the average per-frame MSE of a single train and test rollout for each dataset shown in Fig. 6.

Note that the different versions of HGN have different trade-offs. The deterministic version produces more accurate reconstructions but it does not allow sampling. This effect is equivalent to a similar distinction between autoencoders (Hinton & Salakhutdinov, 2006) and VAEs. Using the simpler Euler integrator rather than the more involved leapfrog one might be conceptually more appealing, however it does not provide the same energy conservation and reversibility properties as the leapfrog integrator, as evidenced by the increase by an order of magnitude of the variance of the learned Hamiltonian throughout a sequence rollout as shown in Tbl. 2. The full version of HGN, on the other hand, is capable of reproducing the dynamics well, is capable of producing diverse yet plausible rollout samples (Fig. 8) and its rollouts can be reversed in time, sped up or slowed down by either changing the value or the sign of $dt$ used in the integrator (Fig. 7).

**Expressive density modelling using learned Hamiltonian flows**    We evaluate whether NHF is capable of expressive density modelling by stacking learned Hamiltonians into a series of normalising

| MODEL | MASS-SPRING | | PENDULUM | | TWO-BODY | | THREE-BODY | |
| --- | --- | --- | --- | --- | --- | --- | --- | --- |
| | TRAIN | TEST | TRAIN | TEST | TRAIN | TEST | TRAIN | TEST |
| HNN | N/A | N/A | N/A | N/A | N/A | N/A | 0.72 | 1.43 |
| HNN (CONV) | N/A | N/A | N/A | N/A | N/A | N/A | 2.17 | 28.34 |
| HGN (EULER) | 157.28 | 120.16 | 135.53 | 19.53 | 405.83 | 682.02 | 1.18 | 8.36 |
| HGN (DETERM) | 0.08 | 0.09 | 0.03 | 0.02 | 0.03 | 0.05 | 3.08 | 0.22 |
| HGN (NO $f_\psi$) | 0.02 | 0.02 | 0.02 | 0.04 | 0.04 | 0.02 | 4.03 | 1.82 |
| HGN (LEAPFROG) | 1.38 | 0.68 | 3.15 | 7.13 | 0.15 | 0.45 | 0.31 | 1.65 |

Table 2: Variance of the Hamiltonian on four physical systems over single train and test rollouts shown in Fig. 6. The numbers reported are scaled by a factor of $1e+4$. High variance indicates that the energy is not conserved by the learned Hamiltonian throughout the rollout. Many HNN Hamiltonians have collapsed to 0, as indicated by N/A. HGN Hamiltonians are meaningful, and different versions of HGN conserve energy to varying degrees.

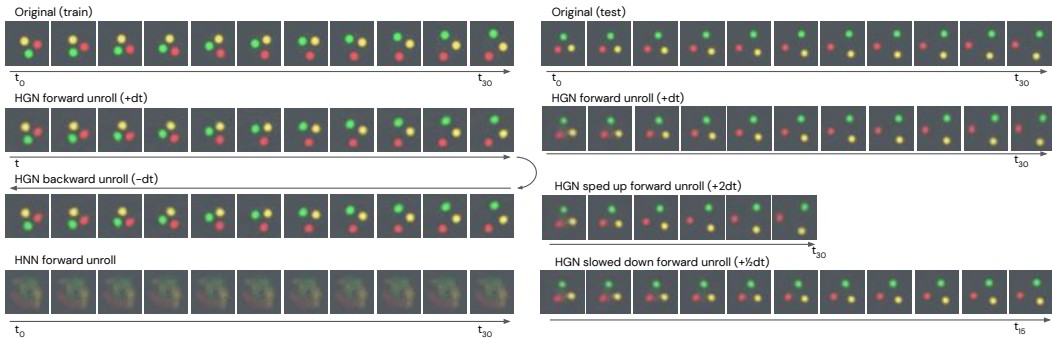

Figure 7: Example of a train and a test sequence from the dataset of a three-body system, its inferred forward, backward, double speed and half speed rollouts in time from HGN, and a forward rollout from HNN. HNN did not learn the dynamics of the system and instead learned to reconstruct an average image.

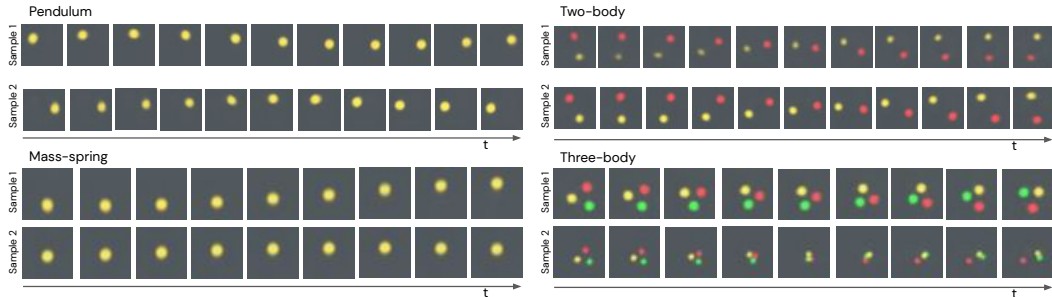

Figure 8: Examples of sample rollouts for all four datasets from a trained HGN.

flows. Fig. 9 demonstrates that NHF can transform a simple soft-uniform prior distribution $\pi(\boldsymbol{s}_0; \sigma, \beta)$ into significantly more complex densities with arbitrary numbers of modes. The soft-uniform density, $\pi(\boldsymbol{s}_0; \sigma, \beta) \propto f(\beta(s + \sigma\frac{1}{2}))f(-\beta(s - \sigma\frac{1}{2}))$, where $f$ is the sigmoid function and $\beta$ is a constant, was chosen to make it easier to visualise the learned attractors. The model also performed well with other priors, including a Normal distribution. It is interesting to note that the trained model is very interpretable. When decomposed into the equivalents of the kinetic and potential energies, it can be seen that the learned potential energy $V(\boldsymbol{q})$ learned to have several local minima, one for each mode of the data. As a consequence, the trajectory of the initial samples through the flow has attractors at the modes of the data. We have also compared the performance of NHF to that of the RNVP baseline

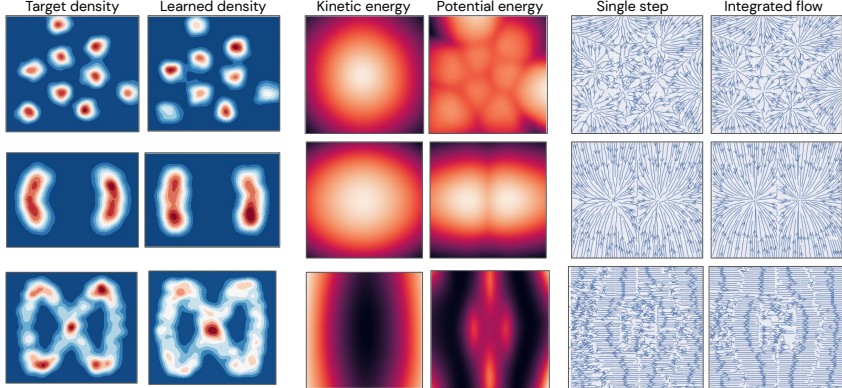

Figure 9: Multimodal density learning using Hamiltonian flows. From left to right: KDE estimators of the target and learned densities; learned kinetic energy $K(\boldsymbol{p})$ and potential energy $V(\boldsymbol{q})$; single leapfrog step and an integrated flow. The potential energy learned multiple attractors, also clearly visible in the integrated flow plot. The basins of attraction are centred at the modes of the data.

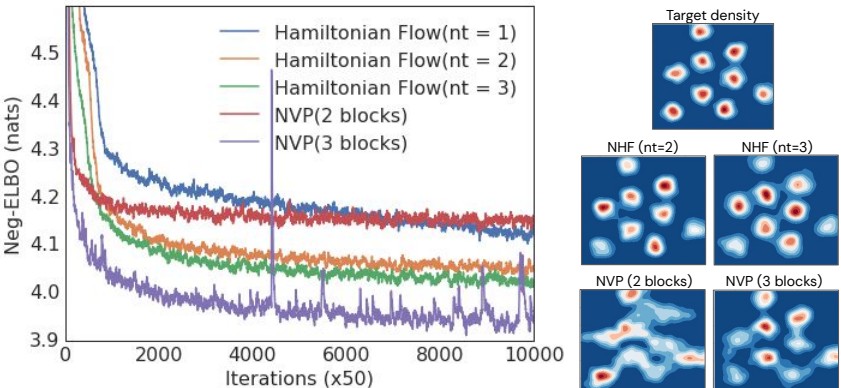

Figure 10: Comparison between RNVP (Dinh et al., 2017) and NHF on a Gaussian Mixture Dataset. The RNVPs use alternating masks with two layers (red) or 3 layers (purple). NHF uses 1, 2 or 3 leapfrog steps (blue, yellow and green respectively).

(Dinh et al., 2017). Fig. 10 shows that the two approaches are comparable in their performance, but NHF is more computationally efficient as discussed at the end of Sec. 3.3.

## 5 CONCLUSIONS

We have presented HGN, the first deep learning approach capable of reliably learning Hamiltonian dynamics from pixel observations. We have evaluated our approach on four classical physical systems and demonstrated that it outperformed the only relevant baseline by a large margin. Hamiltonian dynamics have a number of useful properties that can be exploited more widely by the machine learning community. For example, the Hamiltonian induces a smooth manifold and a vector field in the abstract phase space along which certain physical quantities are conserved. The time evolution along these paths is also completely reversible. These properties can have wide implications in such areas of machine learning as reinforcement learning, representation learning and generative modelling. We have demonstrated the first step towards applying the learnt Hamiltonian dynamics as normalising flows for expressive yet computationally efficient density modelling. We hope that this work serves as the first step towards a wider adoption of the rich body of physics literature around the Hamiltonian principles in the machine learning community.

ACKNOWLEDGEMENTS

We thank Alvaro Sanchez for useful feedback.

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

# A  SUPPLEMENTARY MATERIALS

## A.1  HAMILTONIAN GENERATIVE NETWORK

The Hamiltonian Generative Network (HGN) consists of three major parts, an encoder, the Hamiltonian transition network and a decoder. During training the encoder starts with a sequence

$$(\boldsymbol{x}_1,\boldsymbol{x}_2,...,\boldsymbol{x}_n)\in\mathbb{R}^{32\times32\times3} \tag{7}$$

of raw training images and encodes it into a probabilistic prior representation transformed with an additional network on top

$$q=q(\boldsymbol{z}|\boldsymbol{x}_1,\boldsymbol{x}_2,...,\boldsymbol{x}_n)\sim p=\mathcal{N}(\boldsymbol{s}_0|(0,1)) \tag{8}$$

into a start state

$$\boldsymbol{s}_0\in\mathbb{R}^{4\times4\times(2\times16)} \tag{9}$$

consisting of a downsized spatial representation in latent space ($4\times4$), where each abstract pixel is the the concatenation of abstract position ($\boldsymbol{q}$) and momentum ($\boldsymbol{p}$) (each of dimension 16). The encoder network is a convolutional neural network with 8 layers, with 32 filters on the first layer, then 64 filters on each subsequent layer, while in the last layer we have 48 filters. The final encoder transformer network is a convolutional neural network with 3 layers and 64 filters on each layer. Starting from this initial embedded state, the Hamiltonian transition network generates subsequent states using a symplectic integrator approximating the Hamiltonian equations. The Hamiltonian transition network represents the Hamiltonian function

$$\mathcal{H}:\boldsymbol{s}_t\in\mathbb{R}^{4\times4\times(2\times16)}\to\mathbb{R} \tag{10}$$

as a function from the abstract position and momentum space to the real numbers at any time step $t$. The Hamiltonian transition network is a convolutional neural network of 6 layers, each consisting of 64 filters. The discrete timestep we use for the symplectic integrator update step is $dt=0.125$.

At each time step $t$ the decoder network $d_\theta$ takes only the abstract position part $\boldsymbol{q}_t$ of the state $\boldsymbol{s}_t$ and decodes it back to an output image $\widetilde{\boldsymbol{x}}_t\in\mathbb{R}^{32\times32\times3}$ of the same shape as the input images.

$$d_\theta:\boldsymbol{q}_t\to\widetilde{\boldsymbol{x}}_t. \tag{11}$$

The decoder network is a progressive network consisting of 3 residual blocks, where each residual block resizes the current input image by a factor of 2 using the nearest neighbor method (at the end we have to upscale our latent spatial dimension of 4 to the desired output image dimension of 32 in these steps), followed by 2 blocks of a one layer convolutional neural network with 64 filters and a leaky ReLU activation function, closing by a sigmoid activation in each block. After the 3 blocks a final one layer convolutional neural network outputs the output image with the right number of channels.

We use Adam optimisier (Kingma & Ba, 2014) with learning rate 1.5e-4. When optimising the loss, in practice we do not learn the variance of the decoder $p_\theta(\boldsymbol{x}|\boldsymbol{s})$ and fix it to 1, which makes the reconstruction objective equivalent to a scaled L2 loss. Furthermore, we introduce a Lagrange multiplier in front of the KL term and optimise it using the same method as in Rezende & Viola (2018).

## A.2  NEURAL HAMILTONIAN FLOW

For all NHF experiments the Hamiltonian was of the form $\mathcal{H}(q,p) = K(p) + V(q)$. The kinetic energy term $K$ and the potential energy term $V$ are soft-plus MLPs with layer-sizes $[d,128,128,1]$ where $d$ is the dimension of the data. Soft-plus non-linearities were chosen because the optimisation of Hamiltonian flows involves second-order derivatives of the MLPs used for parametrising the Hamiltonians. This makes ReLU non-linearities unsuitable. The encoder network was parametrized as $f_\psi(\boldsymbol{p}|\boldsymbol{q})=\mathcal{N}(\boldsymbol{p};\mu(\boldsymbol{q}),\sigma(\boldsymbol{q}))$, where $\mu$ and $\sigma$ are ReLU MLPs with size $[d,128,128,d]$. The Hamiltonian flow $\mathcal{H}^{dt}$, was approximated using a leapfrog integrator (Neal et al., 2011) since it preserves volume and is invertible for any $dt$ (see also section A.6). We found that only two leapfrog steps where sufficient for all our examples. Parameters were optimised using Adam (Kingma & Ba, 2014) (learning rate 3e-4) and Lagrange multipliers were optimised using the same method as in Rezende & Viola (2018). All

shown kernel density estimate (KDE) plots used 1000 samples and isotropic Gaussian kernel bandwidth of 0.3. For the RNVP baseline (Dinh et al., 2017) we used alternating masks with two or three layers. Each RNVP layer used an affine coupling parametrized by a two-layer relu-MLP that matched those used in the leapfrog.

## A.3 HAMILTONIAN NEURAL NETWORK

The Hamiltonian Neural Network (HNN) (Greydanus et al., 2019) learns a differentiable function $\mathcal{H}(\boldsymbol{q},\boldsymbol{p})$ that maps a system's state in phase space (its position $\boldsymbol{q}$ and momentum $\boldsymbol{p}$) to a scalar quantity interpreted as the system's Hamiltonian. This model is trained so that $\mathcal{H}(\boldsymbol{q},\boldsymbol{p})$ satisfies the Hamiltonian equation by minimizing

$$\mathcal{L}_{\text{HNN}} = \frac{1}{2}[(\frac{\partial\mathcal{H}}{\partial\boldsymbol{p}} - \frac{d\boldsymbol{q}}{dt})^2 + (\frac{\partial\mathcal{H}}{\partial\boldsymbol{q}} + \frac{d\boldsymbol{p}}{dt})^2], \tag{12}$$

where the derivatives $\frac{\partial\mathcal{H}}{\partial\boldsymbol{q}}$ and $\frac{\partial\mathcal{H}}{\partial\boldsymbol{p}}$ are computed by backpropagation. In the original paper, these targets are either assumed to be known or are estimated by finite differences using the state at times $t$ and $t+1$. Accordingly, in the majority of the experiments presented by the authors, the Hamiltonian was learned from the ground truth state space directly, rather than from pixel observations.

The original HNN model is trained in a supervised fashion on the ground truth state of a physical system and its time derivatives. As such, it is not directly comparable to our method, which learns a Hamiltonian directly from pixels. Instead, we compare to the PixelHNN variant of the HNN, which is introduced in the same paper, and which is able to learn a Hamiltonian from images and in the absence of true state or time derivative in some settings.

This required a modification of the model. First, the input to the model became a concatenated, flattened pair of images $X_t = (\boldsymbol{x}_t, \boldsymbol{x}_{t+1})$, which was then mapped to a low-dimensional embedding space $\boldsymbol{z}_t = (\boldsymbol{q}_t, \boldsymbol{p}_t)$ using an encoder neural network. Note that the dimensionality of this embedding ($\boldsymbol{z} \in \mathbb{R}^2$ in the case of the pendulum system presented in the paper) is chosen to perfectly match the ground truth dimensionality of the phase space, which was assumed to be known a priori. This, however, is not always possible, as when a system has not yet been identified. The latent embedding was then treated as an estimate of the position and the momentum of the system depicted in the images, where the momentum was assumed to be equal to the velocity of the system – an assumption enforced by the additional constraint found necessary to encourage learning, which encouraged the time derivative of the position latent to equal the momentum latent using finite differences on the split latents:

$$\mathcal{L}_{\text{CC}} = (\boldsymbol{p}_t - (\boldsymbol{q}_{t+1} - \boldsymbol{q}_t))^2. \tag{13}$$

This loss is motivated by the observation that in simple Newtonian systems with unit mass, the system's state is fully described by the position and its time derivative (the system's velocity). An image embedding that corresponds to the position and velocity of the system will minimize this loss. This assumption is appropriate in the case of the simple pendulum system presented in the paper, however it does not hold more generally.

As mentioned earlier, PixelHNN takes as input a concatenated, flattened pair of images and maps them to an embedding space $\boldsymbol{z}_t = (\boldsymbol{q}_t, \boldsymbol{p}_t)$, which is treated as an estimate of the position and momentum of the system depicted in the images. Note that $X_t$ always consists of two images in order to make the momentum observable. This embedding space is used as the input to an HNN, which is trained to learn the Hamiltonian of the system as before, but using the embedding instead of the true system state.

To enable stable learning in this configuration, the PixelHNN uses a standard mean-squared error autoencoding loss:

$$\mathcal{L}_{\text{AE}} = \frac{1}{N}\sum_{i=1}^{N}(X_t^i - \hat{X}_t^i)^2, \tag{14}$$

where $\hat{X}_t$ is the autoencoder output and $X_t^i$ is the value of pixel $i$ of $N$ total pixels in $X_t$. [2] This loss encourages the network embedding to reflect the content of the input images and to avoid the trivial solution to (12).

The full PixelHNN loss is:

$$\mathcal{L}_{\text{PixelHNN}} = \mathcal{L}_{\text{AE}} + \mathcal{L}_{\text{CC}} + \lambda_{\text{HNN}}\mathcal{L}_{\text{HNN}} + \lambda_{\text{WD}}\mathcal{L}_{\text{WD}}, \tag{15}$$

where $\mathcal{L}_{\text{HNN}}$ is computed using the finite difference estimate of the time derivative of the embedding. $\lambda_{\text{HNN}}$ is a Lagrange multiplier, which is set to 0.1, as in the original paper. $\mathcal{L}_{\text{WD}}$ is a standard L2 weight decay and its Lagrange multiplier $\lambda_{\text{WD}}$ is set to 1e-5, as in the original paper.

In the experiments presented here, we reimplemented the PixelHNN architecture as described in Greydanus et al. (2019) and trained it using the full loss (15). As in the original paper, we used a PixelHNN with HNN, encoder, and decoder subnetworks each parameterized by a multi-layer perceptron (MLP). The encoder and decoder MLPs use ReLU nonlinearities. Each consists of 4 layers, with 200 units in each hidden layer and an embedding of the same size as the true position and momentum of the system depicted (2 for mass-spring and pendulum, 8 for two-body, and 12 for 3-body). The HNN MLP uses tanh nonlinearities and consists of two hidden layers with 200 units and a one-dimensional output.

To ensure the difference in performance between the PixelHNN and HGN are not due primarily to archiectural choices, we also compare to a variant of the PixelHNN architecture using the same convolutional encoder and decoder as used in HGN. We used identical hyperparameters to those described in section A.1. We map between the convolutional latent space used by the encoder and decoder and the vector-valued latent required by the HNN using one additional linear layer for the encoder and decoder.

In the original paper, the PixelHNN model is trained using full-batch gradient descent. To make it more comparable to our approach, we train it here using stochastic gradient descent using minibatches of size 64 and around 15000 training steps. As in the original paper, we train the model using the Adam optimizer (Kingma & Ba, 2014) and a learning rate of 1e-3. As in the original paper, we produce rollouts of the model using a Runge-Kutta integrator (RK4). See Section A.6 for a description of RK4. Note that, as in the original paper, we use the more sophisticated algorithm implemented in scipy (`scipy.integrate.solve_ivp`) (Jones et al., 2001).

## A.4   DATASETS

The datasets for the experiments described in 4 were generated in a similar manner to Greydanus et al. (2019) for comparative purposes. All of the datasets simulate the exact Hamiltonian dynamics of the underlying differential equation using the default scipy initial value problem solver Jones et al. (2001). After creating a dataset of trajectories for each system, we render those into a sequence of images. The system depicted in each dataset can be visualized by rendering circular objects:

- For the mass-spring the mass object is rendered as a circle and the spring and pivot are invisible.
- For the pendulum only the weight (the bob) is rendered as a circle, while the massless rod and pivot are invisible.
- For the two and three body problem we render each point mass as a circle in a different color.

Additionally, we smooth out the circles such that they do not have hard edges, as can be seen in Fig. 7.

## A.5   CONVERGENCE OF HGN AND HNN

In order to obtain the results presented in Tbl. 1 we trained both HGN and HNN for 15000 iterations, with batch size of 16 for HGN, and 64 for the HNN. Given the dataset sizes, this means that HGN was trained for around 5 epochs and HNN was trained for around 19 epochs, which took around 16 hours. Figs. 11-12 plot the convergence rates for HGN (leapfrog) and HNN (conv) on the four datasets.

---

[2]In our experiments, which use RGB images, this expectation is taken over color channels as well as pixels.

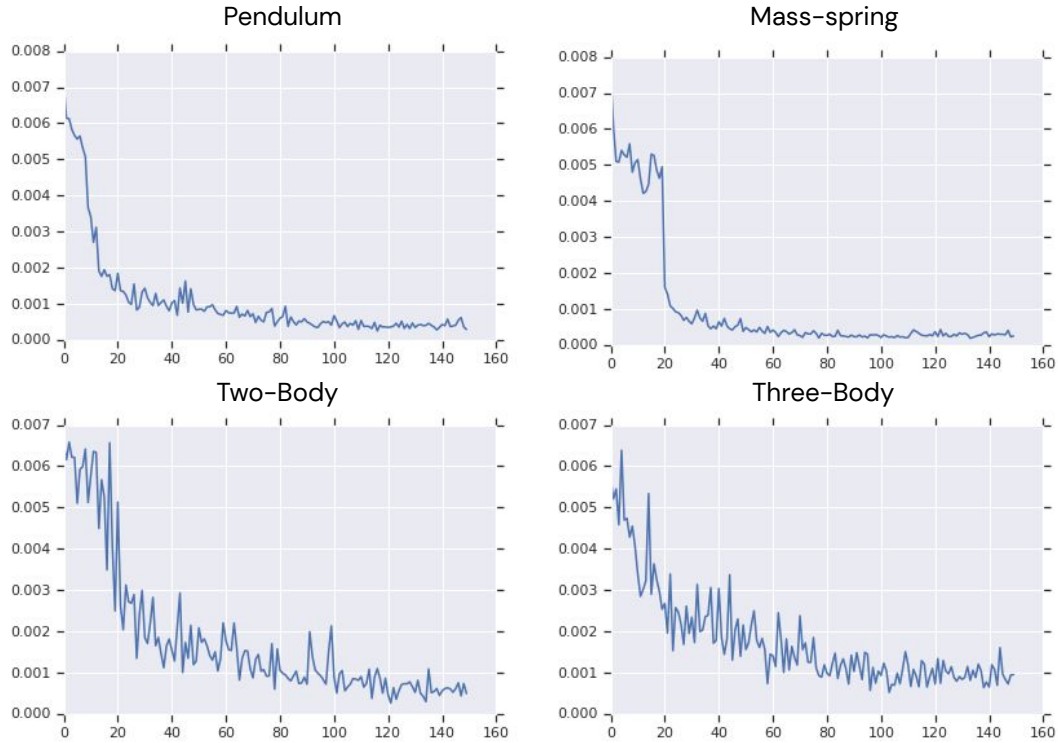

Figure 11: Average MSE for pixel reconstructions during training of HGN (leapfrog). The x axis indicates the training iteration number $1e+2$.

## A.6 INTEGRATORS

Throughout this paper, we estimate the future state of systems from inferred values of the system position and momentum by numerically integrated the Hamiltonian. We explore three methods of numerical integration: (i) Euler integration, (ii) Runge-Kutta integration and (iii) leapfrog integration.

Euler integration estimates the value of a function at time $t+dt$ by incrementing the function's value with the value accumulated by the function's derivative, assuming it stays constant in the interval $[t,t+dt]$. In the Hamiltonian framework, Euler integration takes the form:

$$\boldsymbol{q}_{t+dt} = \boldsymbol{q}_t + dt \frac{\partial \mathcal{H}}{\partial \boldsymbol{p}}\bigg|_{\boldsymbol{p}=\boldsymbol{p}_t} \tag{16}$$

$$\boldsymbol{p}_{t+dt} = \boldsymbol{p}_t - dt \frac{\partial \mathcal{H}}{\partial \boldsymbol{q}}\bigg|_{\boldsymbol{q}=\boldsymbol{q}_t} \tag{17}$$

Because Euler integration estimates a function's future value by extrapolating along its first derivative, the method ignores the contribution of higher-order derivatives to the function's change in time. Accordingly, while Euler integration can reasonably estimate a function's value over short periods, its errors accumulate rapidly as it is integrated over longer periods or when it is applied multiple times. This limitation motivates the use of methods that are stable over more steps and longer integration times.

One such method is four-step Runge-Kutta integration (RK4), the most widely used member of the Runge-Kutta family of integrators. Whereas Euler integration estimates the value of a function at time $t+dt$ using only the function's derivative at time $t$, RK4 accumulates multiple estimates of the function's value in the interval $[t, t+dt]$. This integral more correctly reflects the behavior of the function in the interval, resulting in a more stable estimate of the function's value.
RK4 estimates the state at time $t+dt$ as:

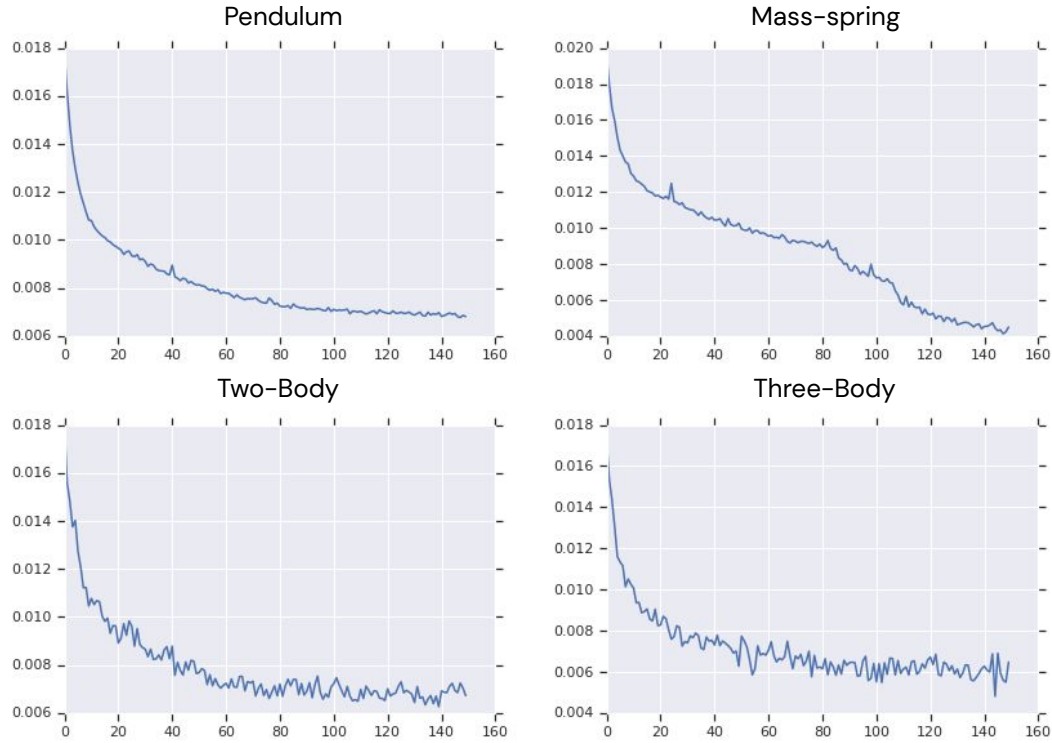

Figure 12: Average MSE for pixel reconstructions during training of HNN (conv). The x axis indicates the training iteration number $1e+2$.

$$\boldsymbol{x}_{t+dt} = \boldsymbol{x}_t + dt(\frac{1}{6}\boldsymbol{k}_1 + \frac{1}{3}\boldsymbol{k}_2 + \frac{1}{3}\boldsymbol{k}_3 + \frac{1}{6}\boldsymbol{k}_4), \tag{18}$$

where, for compactness, we write the full system state as $\boldsymbol{x}_t = [\boldsymbol{q}_t, \boldsymbol{p}_t]$ and the Hamiltonian as $\mathcal{H}(\boldsymbol{x})$. In the Hamiltonian framework, $\boldsymbol{k}_1, \boldsymbol{k}_2, \boldsymbol{k}_3, \boldsymbol{k}_4$ are obtained by evaluating the derivative at four points in the interval $[t, t+dt]$:

$$\boldsymbol{k}_1 = \left. \frac{d\mathcal{H}(\boldsymbol{x})}{dt} \right|_{\boldsymbol{x}=\boldsymbol{x}_t} \tag{19}$$

$$\boldsymbol{k}_2 = \left. \frac{d\mathcal{H}(\boldsymbol{x})}{dt} \right|_{\boldsymbol{x}=\boldsymbol{x}_t + \frac{dt}{2}\boldsymbol{k}_1} \tag{20}$$

$$\boldsymbol{k}_3 = \left. \frac{d\mathcal{H}(\boldsymbol{x})}{dt} \right|_{\boldsymbol{x}=\boldsymbol{x}_t + \frac{dt}{2}\boldsymbol{k}_2} \tag{21}$$

$$\boldsymbol{k}_4 = \left. \frac{d\mathcal{H}(\boldsymbol{x})}{dt} \right|_{\boldsymbol{x}=\boldsymbol{x}_t + dt \cdot \boldsymbol{k}_3} \tag{22}$$

While RK4 may produce reasonably stable estimates over short periods of time, it is not guaranteed to behave stably indefinitely. Neither RK4 nor Euler integration is guaranteed to preserve the energy of the system being integrated, and in practice both will produce estimates that drift away from the true system dynamics over timescales that are relevant for simulating real systems.

Fortunately, there are well-known methods for numerical integration that preserve energy and can be applied to Hamiltonian systems, like the one we propose here. One such method is leapfrog integration, which is a special method for integrating differential equations of the form:

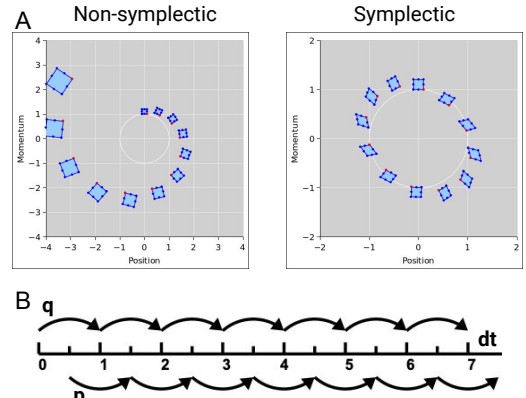

Figure 13: **A**: example of using a symplectic (leapfrog) and a non-symplectic (Euler) integrators on the Hamiltonian of a harmonic oscillator. The blue quadrilaterals depict a volume in phase space over the course of integration. While the symplectic integrator conserves the volume of this region, but the non-symplectic integrator causes it to increase in volume with each integration step. The symplectic integrator clearly introduces less divergence in the phase space than the non-symplectic alternative over the same integration window. **B**: an illustration of the leapfrog updates in the phase space, where $q$ is position and $p$ is momentum.

$$\frac{dy}{dt} = F(x), \quad \frac{dx}{dt} = y. \tag{23}$$

If we assume that the Hamiltonian equations take this form, we can integrate them using the leapfrog integrator, which in essence updates the position and momentum variables at interleaved time points in a way that resembles the updates "leapfrogging" over each other (see Fig. 13B for an illustration).

In particular, the following updates can be applied to a Hamiltonian of the form $\mathcal{H} = V(q) + T(p)$, where $V$ is the potential energy and $T$ is the kinetic energy of the system:

$$\boldsymbol{p}_{t+dt\frac{1}{2}} = \boldsymbol{p}_t - \frac{dt}{2} \frac{\partial V}{\partial \boldsymbol{q}}\bigg|_{\boldsymbol{q}=\boldsymbol{q}_t} \tag{24}$$

$$\boldsymbol{q}_{t+dt} = \boldsymbol{q}_t + dt\, \boldsymbol{p}_{t+\frac{dt}{2}}. \tag{25}$$

As discussed above, leapfrog integration is more stable and accurate over long rollouts than integrators like Euler or RK4. This is because the leapfrog integrator is an example of a *symplectic* integrator, which means it is guaranteed to preserve the special form of the Hamiltonian even after repeated application. An example visual comparison between a symplectic (leapfrog) and non-symplectic (Euler) integrator applied over the Hamiltonian for a harmonic oscilator is shown in Fig. 13A. For a more thorough discussion of the properties of leapfrog integration, see (Neal et al., 2011).

### A.7 PROOF OF THE HAMILTONIAN FLOW ELBO

The Hamiltonian Flow consists of two components. Firstly, it defines a density model over the joint space $s_T = (q_T, p_T)$ using the Hamiltonian Flow as described in section 3.3. However, we assume that the observable variable represents only $q_T$ and treat $p_T$ as a latent variable which we have to marginalize over. Since the integral is intractable using the introduced variational distribution we can derive the lower bound on the marginal likelihood:

$$
\begin{aligned}
\ln p(\boldsymbol{q}_T) &= \ln \int p(\boldsymbol{q}_T, \boldsymbol{p}_T)\, d\boldsymbol{p}_T \\
&= \ln \int \frac{p(\boldsymbol{q}_T, \boldsymbol{p}_T)}{f_\psi(\boldsymbol{p}_T|\boldsymbol{q}_T)} f_\psi(\boldsymbol{p}_T|\boldsymbol{q}_T)\, d\boldsymbol{p}_T \\
&= \ln \mathbb{E}_{f_\psi(\boldsymbol{p}_T|\boldsymbol{q}_T)} \Big[ \frac{p(\boldsymbol{q}_T, \boldsymbol{p}_T) d\boldsymbol{p}_T}{f_\psi(\boldsymbol{p}_T|\boldsymbol{q}_T)} \Big] \\
&\geq \mathbb{E}_{f_\psi(\boldsymbol{p}_T|\boldsymbol{q}_T)} \Big[ \ln \frac{p(\boldsymbol{q}_T, \boldsymbol{p}_T) d\boldsymbol{p}_T}{f_\psi(\boldsymbol{p}_T|\boldsymbol{q}_T)} \Big] \\
&= \mathbb{E}_{f_\psi(\boldsymbol{p}_T|\boldsymbol{q}_T)} \Big[ \ln \pi (\mathcal{H}_1^{-dt} \circ ... \circ \mathcal{H}_T^{-dt}(\boldsymbol{q}_T, \boldsymbol{p}_T)) - \ln f_\psi(\boldsymbol{p}_T|\boldsymbol{q}_T) \Big] \\
&= \mathrm{ELBO}(\boldsymbol{q}_T)
\end{aligned}
$$

