# OpenReview forum: "Hamiltonian Generative Networks"
_ICLR.cc/2020/Conference — Accept (Spotlight)_

### Official Review · AnonReviewer2 · 2019-10-11
**Official Blind Review #2**

**Rating:** 6

**Review:**

The paper proposes two ideas: 1) Hamiltonian Generative Networks (HGN) and 2) Neural Hamiltonian Flow (NHF).

Hamiltonian Generative Networks are generative models of high-dimensional timeseries which use hamiltonian differential equations to evolve latent variables (~position and ~momentum vectors) through time (using any differentiable integration scheme). Given the ~position vector a decoder network generate predictions. The initial latent variables are inferred using a VAE style inference network that takes a sequence of images as the input. The decoder, inference network and crucially the hamiltonian energy function is learned by minimizing a VAE style ELBO on observed sequences. The model induces a strong hamiltonian physics prior and is quite elegant all in all. The model is evaluated on 4 simulated physics tasks, and beats the only baseline the Hamiltonian Neural Network (HNN).

Neural Hamiltonian Flow notes that hamiltonian dynamics are invertible and volume preserving, which is the properties you need for neural flow models. As such it propose to use a series of hamiltonian update steps with multiple learned energy functions as a flexible density estimator. The resulting density estimator is subjectively evaluated on three 2d toy density estimation tasks.

I propose a weak accept as I think the paper is interesting and well written, but could be much better. The paper explains how both HGN and NHF work, but not much more. The HGN is only compared to a single other method (the closely related HNN), on four toy benchmarks. The NHF is barely evaluated, and not compared to anything.

Does the authors actually care about modelling physics and think their method is superior at this? If so, they should compare and contrast to some of the many, many papers on modelling physics, e.g. [1,2,3,4] and references herein. If not, what do they care about? Where do they think this model can be useful? Why should anyone use this model over some of the many, many other models one could use?

Similarly for the NHF, if I only read this paper I have no idea whether it's better than any of the other flow based models. Is it faster (to sample? to eval likelihood?) is it a better estimator? Why should I use it?

I think the paper would benefit from being split into two papers, each thoroughly examining one idea.

A few questions and minor comments

 - While the hamiltonian dynamics expect position and momentum vectors, the neural network is free to use those however it sees fit. Actually, if I understand correctly, the position vector must also encode the color of the objects for the 2 and 3 body problem. Is that correct? It would be interesting if you could examine how predictive the q and p vectors were of the true position and momentum vectors.
 - Successful experiments with n-body problems with n randomly sampled during training and unseen n used in testing would be very powerful in showing generalization. I'm afraid that the current setup doesn't generalize well.
 - I'm surprised that the generated images start showing artifacts after some time, e.g. pendulum sample 4 and 6 in https://docs.google.com/presentation/d/e/2PACX-1vRD2FnKgymgR2lU8lE6-XM8Cz-UWLTI6n_Uht3v6Gu4hIyMHmOcNL5D-0eG6Z4WHDAWS4qFosU-lxXP/pub?start=false&loop=false&delayms=3000&slide=id.g61bbdf339d_0_426. How can those appear if the hamiltonian dynamics preserve energy?
 - Equation 3 is given as self evident. It's not clear to me why 1) det(I+dt*A) = 1+dt*Tr(A)+O(dt^2). Can the authors give a reference? Also, doesn't the O(dt^2) term accumulate over multiple timesteps or longer rollouts? if so, how can the multiple steps proposed be said to be volume preserving?

[1] - Battaglia, Peter, et al. "Interaction networks for learning about objects, relations and physics." Advances in neural information processing systems. 2016.
[2] - de Avila Belbute-Peres, Filipe, et al. "End-to-end differentiable physics for learning and control." Advances in Neural Information Processing Systems. 2018.
[3] - Santoro, Adam, et al. "A simple neural network module for relational reasoning." Advances in neural information processing systems. 2017.
[4] - Fraccaro, Marco, et al. "A disentangled recognition and nonlinear dynamics model for unsupervised learning." Advances in Neural Information Processing Systems. 2017.

**Experience Assessment:**

I have published one or two papers in this area.

**Review Assessment: Checking Correctness Of Derivations And Theory:**

I assessed the sensibility of the derivations and theory.

**Review Assessment: Checking Correctness Of Experiments:**

I assessed the sensibility of the experiments.

**Review Assessment: Thoroughness In Paper Reading:**

I made a quick assessment of this paper.

---

> ### Author Response · Authors · 2019-11-12
> **Response (part 1)**
>
> Dear Reviewer, thank you for your thoughtful comments and feedback.
>
> [Why is Hamiltonian flow more efficient]: The benefits of the NHF over standard flows, as discussed at the end of section 3.3, are that since the NHF is volume preserving by design we do not need to calculate the standard log determinant of the likelihood, because it is 0. Additionally, similar to Neural ODEs (and RevNets for instance) you do not need to store the forward pass for computing gradients as the model can be directly inverted. Finally, we can even use simple symplectic integrators, like the leapfrog integrator, when the Hamiltonian is separable, and this defines a valid volume preserving transformation even in the discrete case (e.g. not only in the limit dt->0). We are also running extra experiments to produce a quantitative comparison between NHF and some of the existing alternatives. We will include these results in the paper as soon as they are ready.
>
> [Relevance of the model in the context of past work]: Thank you for raising this point and for providing these references. As described in the introduction to our submission, the main goal of our paper is to introduce the Hamiltonian formalism to contemporary machine learning, since it plays a central role in physics, the theory and tools around it have a long history of development and use, and it has properties that many machine learning systems may benefit from, like time reversibility, smooth interpolation in time and conservation of state properties. Given that this is a relatively new line of work, we start by modelling image sequences with well understood underlying physical dynamics that allow us to evaluate our approach easily. The  aim of this work is not primarily image sequence modelling. This is the reason why we do not compare to the large body of literature that uses neural networks to model image sequences, and instead only focus on the most relevant work with the same motivation as us, the Greydanus et al (2019) paper.
>
> That said, the references you raise are certainly very interesting, despite not being directly comparable to our present work. Of these four papers, de Avila Belbute-Peres et al [2] is the closest in flavour to our work, as it proposes a model that works from images and that models dynamics. [2] uses an autoencoder to map images to a latent variable that can be stepped forward in time using a learned physics model. As such, this model is similar in both design philosophy and architecture to the pixel version of HNN (Greydanus et al, 2019), which, however, is a more direct baseline and which our model consistently outperforms. Moreover, the model in [2] is trained in a supervised or semi-supervised regime, which makes it difficult to baseline against our model, which is unsupervised. We agree that this paper provides important context for understanding our contribution, and we have added it to the revised manuscript.
>
> While Fraccaro et al [4] models image dynamics, it does not focus on modeling systems with unknown Hamiltonian dynamics. The other two papers are less directly relevant: Battaglia et al [1] operates directly from state and leverages domain-specific knowledge about objects, and as such is not directly comparable to our work, which uses images as input and does not use knowledge about the number or structure of objects in the scene. Santoro et al [3] is a model of relational reasoning, and is not designed to model physical system dynamics. The experiments on physical system included probe the model's ability to extract relational information (e.g. classifying the connectivity structure of a set of moving balls), not to model dynamics.

---

> > ### Author Response · Authors · 2019-11-12
> > **Response (part 2)**
> >
> > [Minor questions]
> >
> > [Information in the state space] You are right that the q part of our state space is expected to represent more than just the position of the mass and must also incorporate non-physical properties, such as the colour of the mass. Thank you for suggesting that we explicitly check how predictive the inferred state is of the real p and q values. We are running the relevant experiments and will report the results as soon as they are available.
> >
> > [Generalisation to different numbers of objects] While we do not expect the current implementation of the model to generalise to unseen numbers of objects, we believe that this can be easily addressed by augmenting the architecture with a graph neural network, similarly to the recent paper by Sanchez-Gonzalez et al, 2019. We have, however, tested the generalisation of our model to unseen initial states. We can see that the performance of the model is comparable between the training and test distributions of initial states as reported in Tbl. 1 and Fig. 6.
> >
> > [Artifacts in samples] We agree that in an ideal world we would not expect to see the artifacts that appear in some of our samples. However, given that the reconstruction quality depends not only on the position in phase space, but also on the quality of the decoder network, it is not clear to us what the source of these artifacts is. We plan to investigate this in future work.
> >
> > [Equation 3] The derivative of the determinant is given by Jacobi's formula (see https://en.wikipedia.org/wiki/Jacobi%27s_formula), which simplifies at the identity to: the derivative of the determinant is the trace. We have added a mention of Jacobi's formula in the paper to help the reader. Regarding the term in $dt^2$, our network integrates a differentiable equation, and our formula shows that exact integration of this differentiable equation does indeed created a volume preserving flow. As usual with numerical integration, numerical errors can accumulate over time, although they will be small since they are in the order of $dt^2$.

---

> > > ### Comment · AnonReviewer2 · 2019-11-14
> > > **Update**
> > >
> > > Thank you for your thoughtful comments.
> > >
> > > I appreciate the addition of the quantitative experiments on NHF.
> > >
> > > I've decided to retain my original rating.

---

### Official Review · AnonReviewer3 · 2019-10-23
**Official Blind Review #3**

**Rating:** 6

**Review:**

Summary: The authors present a method for learning Hamiltonian functions that govern a dynamical directly from observational data.  The basic approach uses three networks: 1) an inference network (I'm not clear why this is not just called an encoder), that maps past observations to a latent p,q space in a VAE-like fashion; 2) a Hamiltonian network that governs the time-evolution of the system over this latent state; and 3) a decoder network that outputs the observation from the latent state.  In addition to introducing this basic formalism, the

Comments: I have mixed opinions on this paper, though am leaning slightly toward acceptance.  The overall notion of learning a Hamiltonian network directly is a great one, though really this is due to the Hamiltonian Neural Networks paper of Greydanus et al., 2019.  Although the focus in that work is on applying learned Hamiltonian networks directly to physics-based data, they also have an encoder-decoder network just using a classical autoencoder instead of a VAE.  So my first impression is that the benefits of the proposed HGN over HNNs in Figure 6 is really just an artific of this replacement.

Perhaps because the authors also felt this was a marginal contribution, the paper's ultimate value may prove to be in the consideration of such networks for the purposes of normalizing flow models.  This portion seemed a little bit underdeveloped in the paper, to be honest, but overall the idea of parameterizing a normalizing flow with a Hamiltonian dynamical system seems like a good one (e.g., allowing for easier large-timestep inference).  But on the flipside, it does seem like the presentation here is rather brief, i.e., just defining the ELBO without much context or detail, etc.

Thus, while I'm very much on the fence on this paper, I think the marginal improvement over HNNs via a better encoder/decoder model, plus the realization that these methods are a good fit for normalizing flow models, altogether put the paper slightly above bar for me.

**Experience Assessment:**

I have published one or two papers in this area.

**Review Assessment: Checking Correctness Of Derivations And Theory:**

I assessed the sensibility of the derivations and theory.

**Review Assessment: Checking Correctness Of Experiments:**

I assessed the sensibility of the experiments.

**Review Assessment: Thoroughness In Paper Reading:**

I read the paper thoroughly.

---

> ### Author Response · Authors · 2019-11-12
> **Response**
>
> Dear Reviewer, thank you for your thoughtful comments and feedback.
>
> You are right that our work is closely related to the Hamiltonian Neural Network (HNN) model by Greydanus et al, 2019. We would like to note that our work was done concurrently to the work by Greydanus and colleagues. We would also like to clarify the differences between our Hamiltonian Generative Network (HGN) and HNN. We apologize if this was not made clear in the original manuscript.
>
> The most obvious difference is that HGN is a generative model and uses the VAE framework rather than a deterministic autoencoder to infer the phase space from pixels. However, this difference does not account for our improved results, as even the deterministic version of our model greatly outperforms the HNN (see Table 1 in the original manuscript). The key difference between the two methods is that our proposed HGN learns the Hamiltonian function directly, rather than learning its derivatives. This is a very important distinction. Because the HNN model learns the derivatives of the Hamiltonian, it requires access to the time derivatives dH/dp and dH/dq of the system, which means that the model is trained in an essentially supervised manner. When there is no access to the true derivatives of the Hamiltonian for supervision, one has to resort to finite differences, which can be quite inaccurate. On the other hand, our HGN model learns the energy function directly from data and does not require such supervision. HNN also requires a priori knowledge of the dimensionality of the phase space and a separable Hamiltonian, while our model requires no a priori knowledge and does not depend on a separability assumption. Finally, given that our model is generative, it can be sampled to generate novel rollouts, which HNN cannot. Taking all of these factors together, our model is able to model all of the visual datasets considered in our paper well, while HNN struggles to learn anything beyond an average pixel reconstruction (see see Tbl. 1 and Figs. 6-7 in the original manuscript).
>
> In terms of the Hamiltonian flows part of the paper, we have provided a more detailed proof of the ELBO in Eq. 4 (which is now Eq.6 in the new manuscript) at the end of the Appendix. We are also running extra experiments to produce a quantitative comparison between NHF and some of the existing alternatives, which we will include in the paper as soon as the results are ready.

---

> > ### Comment · AnonReviewer3 · 2019-11-14
> > **A few comments**
> >
> > Thanks for clarifying these points.  I'm still not convinced by the substantial contribution of the VAE versus the deterministic AE that the HNN uses.
> >
> > I do admit that I didn't fully appreciate the point about the training on derivatives versus training the function directly.  This is indeed a reasonable distinction that makes the methods more separate, though it seems to be a fairly straightforward (if indeed important) difference.
> >
> > I'm inclined to keep my score as it is, since (even given the fact that the work was done independently), the work _is_ nonetheless quite similar to the HNN framework.  But as specified, I'm still in favor of accepting the paper.

---

### Official Review · AnonReviewer1 · 2019-10-24
**Official Blind Review #1**

**Rating:** 8

**Review:**

The paper introduces a novel way of learning Hamiltonian dynamics with a generative network. The Hamiltonian generative network (HGN) learns the dynamics directly from data by embedding observations in a latent space, which is then transformed into a phase space describing the system's initial (abstract) position and momentum. Using a second network, the Hamiltonian network, the position and momentum are reduced to a scalar, interpreted as the Hamiltonian of the system, which can then be used to do rollouts in the phase space using techniques known from, e.g., Hamiltonian Monte Carlo sampling. Finally, a decoder network can use the system's phase space location at any rollout step to generate images of the system. The HGN can further be modified, leading to a flow-based model, the Neural Hamiltonian Flow (NHF).
The authors evaluate the HGN on four simulated physical systems, showing substantial improvements over the competing Hamiltonian Neural Network (HNN). Lastly, the NHF is shown to be able to model complex densities, which can further be interpreted in terms of the kinetic and potential energy.

This paper is a rare treasure. It tackles a well-motivated problem and introduces a, to my knowledge, completely new framework for embedding Hamiltonian dynamics in a generative model. This is hugely inspiring! The paper is a joy to read and includes very informative figures providing a high-level understanding of the proposed models. Accept is a no-brainer.

That being said, I have a few questions and suggestions for improvements. My biggest complaint is the evaluation of the NHF model. I would have liked to see a comparison to a state-of-the-art flow-based model in terms of density modelling. The authors state that the NHF offers more expressiveness and computational benefits over standard flow-based models, but this is never shown. While I am willing to believe the claim, it is not intuitive to me, and I would have liked to see experimental verification of it.

Figure 6 needs a bit of love. It is quite challenging to read. Larger font sizes, conversion to vector format, and more distinguishable colours will help a lot.
Additionally, I think it would be helpful to have the derivation of the ELBO in Eq. (4) written out, e.g. in the supplementary material.

Additional questions:
- In the experimental section, I am not sure what is meant by the deterministic version of HGN. Which part if the model is deterministic?
- On p 6, it is mentioned that the Euler integrator results in an increased variance of the learnt Hamiltonian and that this can be seen in Fig. 6. How exactly is this seen in the figure?
- How many epochs were HNN and HGN trained for to produce table 1? How do the convergence rates look, and how long time did they take to train?

Minor comments:
- p 5: Reference to "Salimans et al." is missing the year.
- p 6: There is a hanging ')' after "as shown in Fig. 6)."
- p 6: "reversed it time" -> "reversed in time"
- In the reference for Glow, "Durk P Kingma" should be "Diederik P. Kingma".



**Experience Assessment:**

I have read many papers in this area.

**Review Assessment: Checking Correctness Of Derivations And Theory:**

I assessed the sensibility of the derivations and theory.

**Review Assessment: Checking Correctness Of Experiments:**

I assessed the sensibility of the experiments.

**Review Assessment: Thoroughness In Paper Reading:**

I read the paper at least twice and used my best judgement in assessing the paper.

---

> ### Author Response · Authors · 2019-11-12
> **Response**
>
> Dear Reviewer, thank you for your thoughtful comments and feedback.
>
> [Flows evaluations]: In the paper we do not explicitly claim that the NHF model in necessarily more expressive than other normalizing flows - this by itself is a very difficult comparison to do and it is likely that it is data dependent as well. The key point is that our results suggest that it is at least on par with other flow-based models while providing some computational benefits. These benefits, as discussed at the end of section 3.3 are that since the NHF is volume preserving by design we do not need to calculate the standard log determinant of the likelihood, because it is 0. Additionally, similar to Neural ODEs (and RevNets for instance), you do not need to store the forward pass to compute gradients as the model can be directly inverted. Finally, we can even use simple symplectic integrators, like the leapfrog integrator, when the Hamiltonian is separable, and this defines a valid volume preserving transformation even in the discrete case (e.g. not only in the limit dt->0). We are running extra experiments to produce a quantitative comparison between NHF and some of the existing alternatives, which we will include in the paper as soon as the results are ready.
>
> [Figure 6]: We have updated Fig.6 to improve readability by increasing the resolution of the plots and by splitting the comparisons between HGN, HNN, and the different versions of HGN into separate plots. We have also moved the numbers indicating the variance of the learnt Hamiltonian over a single trajectory into Tbl. 2.
>
> [Derivation of ELBO in Eq. 4]: We have provided a more detailed proof of Eq. 4 (which is now Eq.6 in the new manuscript) at the end of the Appendix. Hopefully, this clears up any confusion about the result.
>
> [Deterministic HGN]: The deterministic version of HGN is equivalent to an autoencoder, where we remove the sampling step from the posterior q(z|x_0...x_T). We have added this clarification to the text.
>
> [Number of epochs and convergence curves]: We trained both models for 15000 iterations, with batch size of 16 for HGN, and 64 for the HNN. This means that HGN trained for around 5 epochs and HNN trained for around 19. This took around 16 hours to run. We have included this information as well as the convergence curves for the four datasets in the Supplementary Materials.
>
> We have also addressed your minor comments and fixed the typos.

---

> > ### Comment · AnonReviewer1 · 2019-11-15
> > **Response to response**
> >
> > Dear authors, thank you very much for your response and the updated paper. Both nicely address my questions.
> >
> > You are right that I was a bit too quick to conclude that you argue that NHF is more expressive than other flow models. I do, however, find the following sentence in the conclusion a bit overselling: "We have demonstrated the first step towards applying the learnt Hamiltonian dynamics as normalising flows for more expressive yet computationally efficient density modelling." To avoid confusion about what you claim, I would suggest simply writing "expressive" rather than "more expressive".
> >
> > I still think this is an excellent paper, although I do agree with reviewer 2 that splitting it in two to allow for a more exhaustive exploration of both models would be good.

---

> > > ### Author Response · Authors · 2019-11-15
> > > **Updated text**
> > >
> > > Thank you for your suggestion. We have updated the text to say "expressive" instead of "more expressive".

---

### Author Response · Authors · 2019-11-13
**Added NHF comparisons with RNVP**

Dear Reviewers,

We have updated the paper with a quantitative comparison between our proposed Neural Hamiltonian Flows and the RNVP (Dinh et al, 2017) baseline. The results are shown in Fig.10 in the updated manuscript.

---

### Decision · Program_Chairs · 2019-12-19

**Decision:**

Accept (Spotlight)

**Comment:**

The paper introduces a novel way of learning Hamiltonian dynamics with a generative network. The Hamiltonian generative network (HGN) learns the dynamics directly from data by embedding observations in a latent space, which is then transformed into a phase space describing the system's initial (abstract) position and momentum. Using a second network, the Hamiltonian network, the position and momentum are reduced to a scalar, interpreted as the Hamiltonian of the system, which can then be used to do rollouts in the phase space using techniques known from, e.g., Hamiltonian Monte Carlo sampling. An important ingredient of the paper is the fact that no access to the derivatives of the Hamiltonian is needed.

The reviewers agree that this paper is a good contribution, and I recommend acceptance.